# A neuromorphic bionic eye with filter-free color vision using hemispherical perovskite nanowire array retina

Zhenghao Long[1,2,3], Xiao Qiu[1,2,3], Chak Lam Jonathan Chan[1], Zhibo Sun[1,2], Zhengnan Yuan[1,2], Swapnadeep Poddar[1,2,3], Yuting Zhang[1], Yucheng Ding[1,2,3], Leilei Gu[4], Yu Zhou[1,2,3], Wenying Tang[1], Abhishek Kumar Srivastava[1,2], Cunjiang Yu[5], Xuming Zou[6], Guozhen Shen[7] & Zhiyong Fan[1,2,3,8]

Spherical geometry, adaptive optics, and highly dense network of neurons bridging the eye with the visual cortex, are the primary features of human eyes which enable wide field-of-view (FoV), low aberration, excellent adaptivity, and preprocessing of perceived visual information. Therefore, fabricating spherical artificial eyes has garnered enormous scientific interest. However, fusing color vision, in-device preprocessing and optical adaptivity into spherical artificial eyes has always been a tremendous challenge. Herein, we demonstrate a bionic eye comprising tunable liquid crystal optics, and a hemispherical neuromorphic retina with filter-free color vision, enabled by wavelength dependent bidirectional synaptic photo-response in a metal-oxide nanotube/perovskite nanowire hybrid structure. Moreover, by tuning the color selectivity with bias, the device can reconstruct full color images. This work demonstrates a unique approach to address the color vision and optical adaptivity issues associated with artificial eyes that can bring them to a new level approaching their biological counterparts.

Cameras for machine vision and robotics are essentially bionic devices mimicking human eyes. These applications require advanced color imaging systems to possess a number of attributes such as high resolution, large FoV, compact design, light-weight and low energy consumption, etc[1]. Conventional imaging systems based on CCD/CMOS image sensors suffer from relatively low FoV, bulkiness, high complexity, and power consumption issues, especially with mechanically tunable optics. Recently, spherical bionic eyes with curved image sensor retinas have triggered enormous research interest[1–7]. This type of devices possess several appealing features such as simplified lens design, low image aberration, wide FoV, and appearance similar to that of the biological eyes rendering them suitable for humanoid robots[8–13]. However, the existing spherical bionic eyes with curved retinas typically only have fixed lens and can only

[1]Department of Electronic and Computer Engineering, The Hong Kong University of Science and Technology, Clear Water Bay, Kowloon, Hong Kong SAR, China. [2]State Key Laboratory of Advanced Displays and Optoelectronics Technologies, HKUST, Clear Water Bay, Kowloon, Hong Kong SAR, China. [3]Guangdong-Hong Kong-Macao Joint Laboratory for Intelligent Micro-Nano Optoelectronic Technology, HKUST, Clear Water Bay, Kowloon, Hong Kong SAR, China. [4]Qingyuan Research Institute, School of Electronic Information and Electrical Engineering, Shanghai Jiao Tong University, No. 800 Dongchuan Road, 200240 Shanghai, China. [5]Department of Engineering Science and Mechanics, Department of Biomedical Engineering, Department of Materials Science and Engineering, Materials Research Institute, Pennsylvania State University, University Park, PA 16802, USA. [6]Key Laboratory for Micro/Nano Optoelectronic Devices of Ministry of Education & Hunan Provincial Key Laboratory of Low-Dimensional Structural Physics and Devices, School of Physics and Electronics, Hunan University, Changsha 410082, China. [7]School of Integrated Circuits and Electronics, Beijing Institute of Technology, Beijing 100081, China. [8]Department of Chemical and Biological Engineering, The Hong Kong University of Science and Technology, Clear Water Bay, Kowloon, Hong Kong SAR, China. ✉e-mail: eezfan@ust.hk

acquire mono color images. Fixed lenses cannot image objects with varying distances. On the other hand, conventional color imaging function of CCD/CMOS image sensors are achieved by using color filter arrays, which add complexity to the device fabrication and cause optical loss[14–19]. Typical absorptive organic dye filters suffer from poor UV and high-temperature stabilities, and plasmonic color filters suffer from low transmission[20–22]. And it is even more challenging to fabricate color filter arrays on hemispherical geometry where most traditional microelectronic fabrication methods are not applicable.

Herein, we demonstrate a novel bionic eye design that possesses adaptive optics and a hemispherical nanowire array retina with filter-free color imaging and neuromorphic preprocessing abilities. The primary optical sensing function of the artificial retina is realized by using a hemispherical all-inorganic $CsPbI_3$ nanowire array that can produce photocurrent without external bias leading to a self-powered working mode. Intriguingly, an electrolyte-assisted color-dependent bidirectional synaptic photo-response is discovered in a well-engineered hybrid nanostructure. Inspired by the vertical alignment of a color-sensitive cone cell and following neurons, the device structure vertically integrates a $SnO_2/NiO$ double-shell nanotube filled with ionic liquid in the core on top of a $CsPbI_3/NiO$ core-shell nanowire. It is found that the positive surrounding gate effect of NiO due to photo hole injection can be partially or fully balanced by electrolyte under shorter (blue) or longer (green and red) wavelength illuminations, respectively. Thus, the device can yield either positive or negative photocurrent under shorter or longer wavelength illumination, respectively. The carriers can be accumulated in $SnO_2/NiO$ structure, giving rise to the bidirectional synaptic photo-response. This color-sensitive bidirectional photo-response instills a unique filter-free color imaging function to the retina. The synaptic behavior-based neuromorphic preprocessing ability, along with the self-powered feature, effectively reduce the energy consumption of the system[23–28]. Moreover, the color selectivity of each pixel can be tuned by a small external bias to detect more accurate color information. We demonstrate that the device can reconstruct color images with high fidelity for convolutional neural network (CNN) classifications. In addition, our bionic eye integrates adaptive optics in the device, by integrating an artificial crystalline lens and an electronic iris based on liquid crystals. The artificial crystalline lens can switch focal length to detect objects from different distances, and the electronic iris can control the amount of light reaching the retina which enhances the dynamic range. Both of the optical components can be easily tuned by the electric field, which are fast, compact, and much more energy efficient compared to the conventional mechanically controlled optics reported hitherto. (Supplementary Table 1 compares our system with some commercial zoom lenses.) The combination of all these unique features makes the bionic eye structurally and functionally equivalent to its biological counterpart.

## Results

### Design of the bionic eye

Figure 1a schematically shows the human retina and Fig. 1b shows the detailed structure of an individual cone cell unit. As suggested, dense photoreceptors arrays, rod and cone cells, are vertically aligned in the hemispherical retina. When an optical stimulus reaches these cells through the cornea and crystalline lens, it is absorbed by these cells to generate neuro-electric pulses. The pulses are subsequently transferred to retina's neural network for preprocessing, where features such as color, brightness, and speed of the moving object are extracted and encoded. The information is then sent to the visual cortex for processing. Under the photoreceptors, the pigment cells can absorb the transmitted light to avoid the blurring from reflection at the bottom. The unique opto-electric conversion and the sensing-computation integration make our retina an efficient data acquisition system that is much more advanced than the current CCD/CMOS image sensors. However, the structure of the eyes can still be further optimized. The inverted structure of retina, which means the signal preprocessing neurons are at the anterior of the photoreceptors, leading to light loss and blind spot issues. And each cone cell is only sensitive to one color, limiting its resolution. Inspired by the

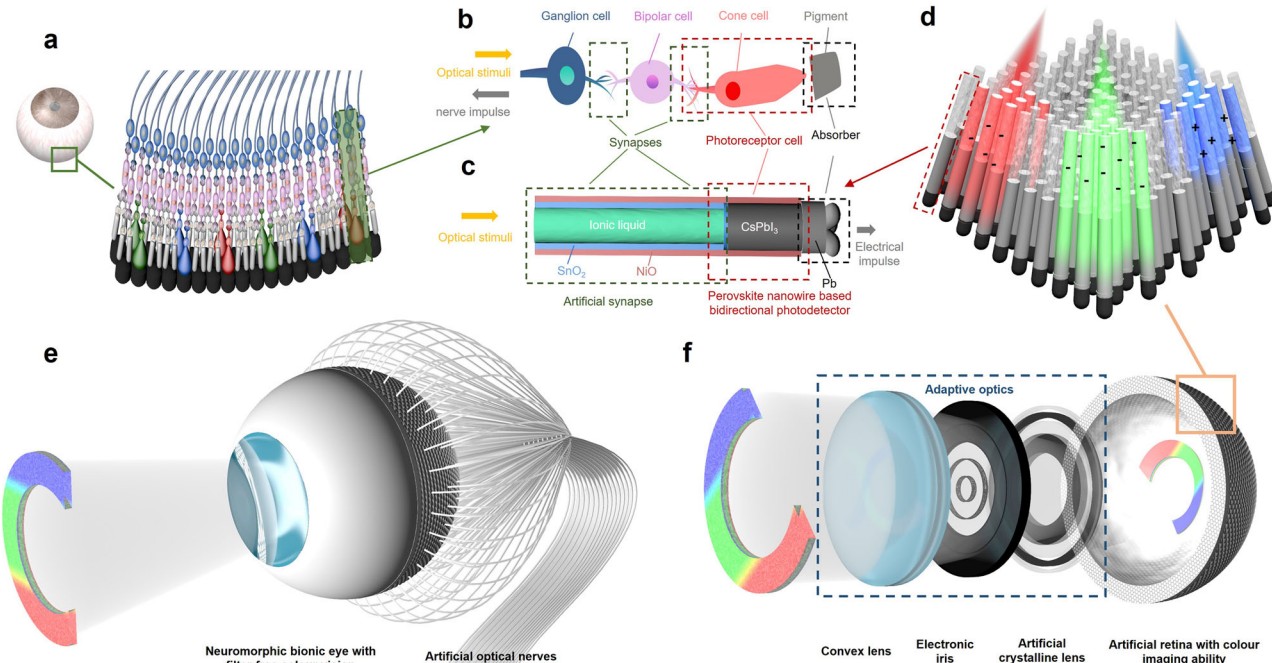

**Fig. 1 | Retina neurons and the neuromorphic bionic eye with filter-free color vision.** Schematics of **a**, human retina neurons, **b**, detailed structure of retina neurons, **c**, nanowire structure, **d**, hemispherical nanowire array with neuromorphic color imaging ability. **e**, The overall structure and **f**, the magnified view of the bionic eye device.

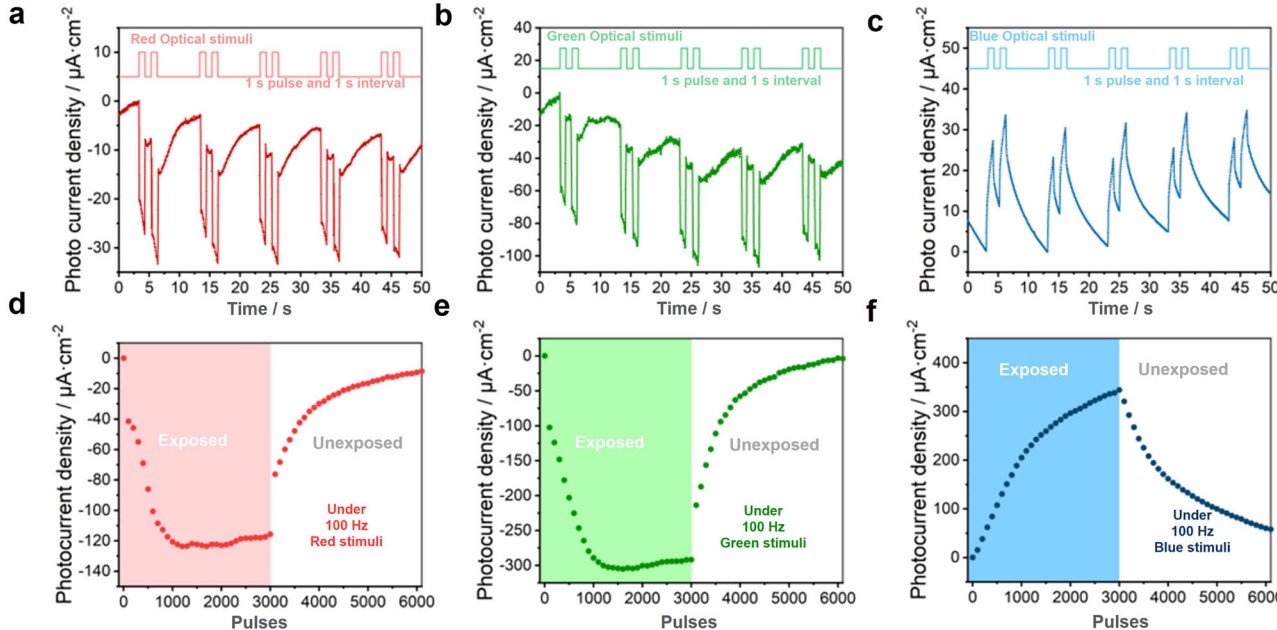

**Fig. 2 | Bidirectional synaptic photo response under 0 V. a–c** Photocurrent density under 11 mW/cm² **a** red (650 nm), **b** green (520 nm), and **c** blue (405 nm) illumination with 1 s pulse width and 1 s pulse interval for 5 cycles under 0 V bias. **d–f** Characterization of bidirectional potentiation and depression process: photocurrent density after certain pulses of 100 Hz and 11 mW/cm² **d** red, **e** green, and **f** blue illumination.

functionalities and delicate design of the human retina, we developed a unique hybrid nanostructure to mimic the structure and function of retina, as schematically shown in Fig. 1c. The nanowire retina fabrication process is shown in Supplementary Fig. 1. In this device structure, a high-density (>10⁹ cm⁻²) hemispherical CsPbI₃ nanowire array (material characterization results are shown in Supplementary Figs. 2 and 3) is used to mimic photoreceptors since CsPbI₃ has a relatively narrow band gap (-1.8 eV) sensitive to the whole visible band. Heterojunctions with metal oxides are designed to modulate the energy bands and enable synaptic bidirectional photo-response behavior, which will be further discussed later in the manuscript. The Pb electrode at bottom can also absorb the remaining light to reduce aberration. The schematic of the hemispherical nanowire array is shown in Fig. 1d. Nanowires are assembled in a hemispherical porous aluminum membrane (PAM), which can structurally support the nanowire array and isolate the adjacent nanowires to avoid cross-talking. Note that the length of the CsPbI₃ nanowire is crucial in determining the photodetection performance. For a -1 μm long NW, under shorter wavelength (e.g., blue light, 0 V bias) illuminations, the carriers are mostly generated on the top of the nanowire and yield positive photocurrent. On the contrary, the device generates negative photocurrent under longer wavelength (green and red light, 0 V bias) illuminations. As such, the device exhibits self-powered neuromorphic color imaging function with a hemispherical geometry. In addition, we have incorporated electrically tunable optics, including an electronic iris and an artificial crystalline lens, aiming to further improve optical adaptivity. The overall structure and the magnified view of our bionic eye are shown in Fig. 1e, f, respectively.

### Pixelated bidirectional synaptic photo-response-based color vision

To achieve low power consumption down to 0 W, we have especially explored the device performance without an external bias voltage. Figure 2a–c shows the self-powered photocurrent under red, green, and blue light pulses for the same light intensity (11 mW cm⁻²). The device yields a positive current under blue light, and a negative current under red/green light. Besides, the current amplitude under green light is higher than that under red light. This effect enables an

intriguing filter-free color recognition functionality. Namely, blue light can be distinguished by current polarity, and red and green light can be distinguished by current amplitude. Apart from the bidirectional photo-response, the device also possesses artificial synaptic plasticity. While the device is exposed to or hidden from illumination, photocurrent gradually elevates and decays, respectively. This behavior shows similarity to the modulation of the strengthening and weakening of the neuronal connections in the human brain. Under two consecutive optical pulses, the response to the second pulse is significantly higher than that to the first pulse, which hints at an artificial paired-pulse facilitation. To further investigate the bidirectional synaptic photo-response, we have characterized the optical potentiation and natural depression of the device. Figure 2d–f shows the photocurrent of the device under 3,000 optical pulses as stimuli with 100 Hz frequency, followed by the depression process in the dark. The gradual enhancement of photocurrent under increasing pulse number paves the way for neuromorphic preprocessing applications.

The color-sensitive, bidirectional synaptic photo-response is a result of the well-engineered 3D nanowire structure. Optoelectronic surrounding gate effect can efficiently tune the photo-response behavior of nanowire structures[29]. In general, holes accumulated in NiO can provide a positive surrounding gate effect to the nanowire. Under shorter and longer wavelengths, the positive charges in NiO can be either fully or partially balanced by the drifting TFSI⁻ ions from the ionic liquid, such that the positive gate effect will be fully or partially suppressed. The difference in the surrounding gate effect results in a reversed photocurrent polarity. Supplementary Fig. 4 shows the schematics of one-unit cell of the hybrid structure and its energy band diagram under shorter and longer wavelength illumination. Supplementary Fig. 4a depicts two carrier transporting paths discussed as follows. Path 1 represents the path for hole transportation along the radial direction of the nanowire following which the holes are injected into NiO and move through NiO shell to reach NiO/SnO₂/Au interface. As the mobility of NiO is relatively low and there is no external bias, the holes may accumulate in the NiO around the CsPbI₃ nanowire leading to a surrounding gate effect. Path 2 represents the axial direction of carrier transport which involves both electrons and holes. Intriguingly, we found that the direction of transport of electrons and holes can be

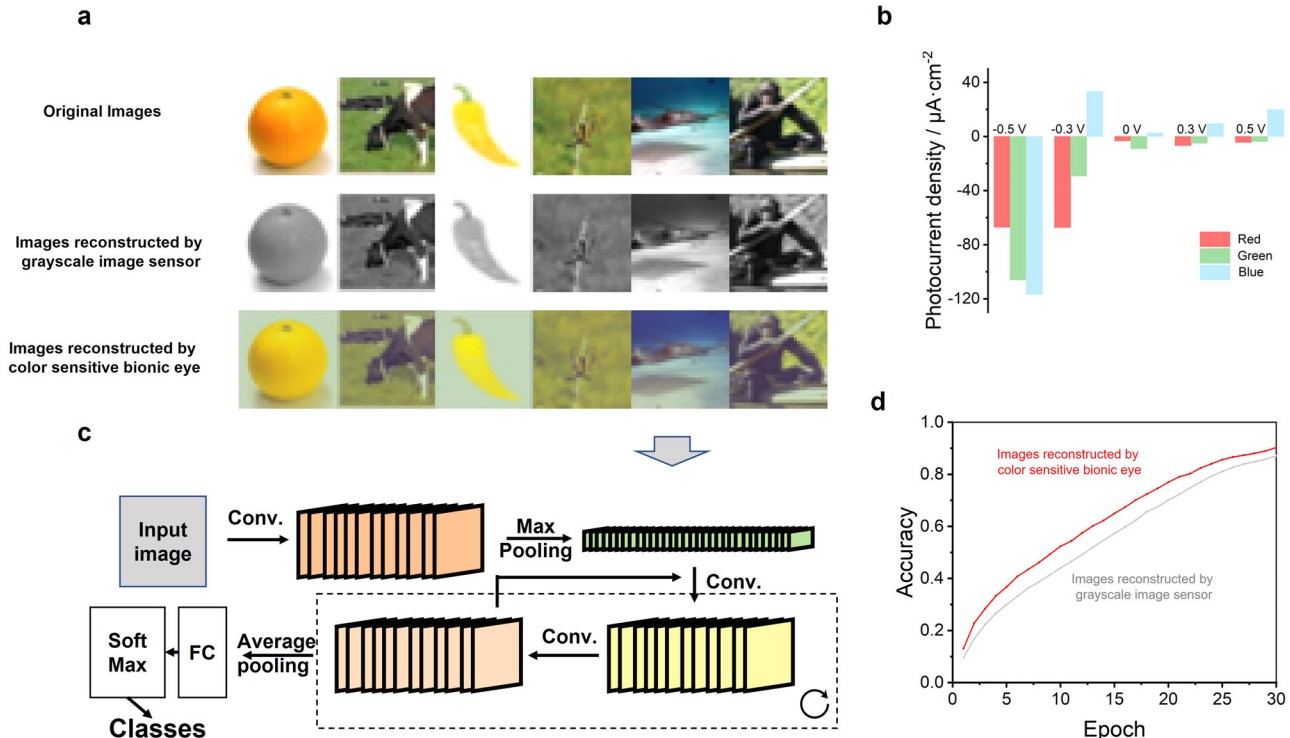

**Fig. 3 | Color image reconstruction and image classification. a** Samples of images from original and reconstructed datasets. **b** photocurrent density of biased artificial retina under red (650 nm, 11 mW/cm²), green (520 nm, 11 mW/cm²), and blue illumination (405 nm, 11 mW/cm²) illuminations. **c** Architecture of the CNN for image classification. **d** Classification accuracy curves of grayscale and full-color reconstructed datasets.

reversed, depending on the magnitude of the aforementioned surrounding gating effect. Specifically, as shown in Supplementary Fig. 4b, blue light has a much shorter penetration depth than that green and red light. Thus, under 405 nm illumination, most carriers are generated on the top surface of the nanowire, near $SnO_2/CsPbI_3$ interface. Supplementary Fig. 4c1 shows schematically the overall carrier transportation under blue light. In path 1, even though holes can be injected to the NiO shell, as they are produced near the $SnO_2/CsPbI_3$ interface, there is no significant amount of holes surrounding the nanowire. The positive charges in NiO can be readily balanced by the negative TFSI⁻ ions in the ionic liquid and thus, the surrounding gate effect to CsPbI3 nanowire is minimal. In the meantime, part of holes can recombine with external electrons emanating from the Au top electrode to generate a positive synaptic photocurrent (Supplementary Fig. 4c2). The accumulation and release of holes result in a gradual change in photocurrent. In path 2 (Supplementary Fig. 4c3), as there is a small barrier at $CsPbI_3/SnO_2$ interface, electrons tend to flow to the bottom Pb electrode. This scenario results in a positive photocurrent. Conversely, under longer wavelength illumination, both electrons and holes are generated deep in the nanowire owing to the larger penetration depth, so that holes injected into NiO shell can wrap around the CsPbI3 nanowires leading to a positive surrounding gate effect, as schematically shown in Supplementary Fig. 4d1, 2. In this case, TFSI⁻ ions can only partially balance the positive charges in NiO shell near $SnO_2/CsPbI_3$ interface. At the $NiO/CsPbI_3$ interface, when a hole injection steady state is reached, the two valance bands are close to a flat band condition. As the nanowire is relatively short, the positive surrounding gate shifts down the energy bands in $CsPbI_3$ with respect to Pb, leading to the formation of a Schottky barrier for electron at $CsPbI_3/Pb$ interface, as shown in Supplementary Fig. 4d3. As such, electrons tend to migrate to $CsPbI_3/SnO_2$ interface and they are collected by $SnO_2$ then transported to Au electrode. And holes tend to move to $CsPbI_3/Pb$ interface to recombine with electrons. This process generates a negative photocurrent measured in the

external circuit, as shown in Supplementary figure 4d3. As there is a Schottky barrier at the $SnO_2/Au$ interface, electrons can accumulate in $SnO_2$ layer and leads to negative synaptic plasticity. Green light with higher photon energy compared to red light, can generate carriers that are more likely to form external current and thus the photocurrent amplitude under green light is higher than that under red light. Moreover, the holes in NiO need to overcome a higher barrier to recombine with external electrons and therefore the device under blue light shows slower photocurrent change and more photocurrent states in potentiation-depression characteristics. We also performed COMSOL simulation of light absorption, electric potential, and density of carriers to study the device working mechanism. The results (Supplementary Fig. 5) firmly support the above discussion. Specifically, light absorption simulation shows red light and green light have longer penetration depth as compared to blue light (Supplementary Fig. 5b1–b3). Holes accumulated in NiO provide a positive surrounding gate that raise the potential of $CsPbI_3$, and lead to the bidirectional photo-response (Supplementary Fig. 5c1–d3). And the relatively higher density of carriers in $CsPbI_3$ NW supports that red and green light can generate carriers deeper within the nanowire compared to that of blue light (Supplementary Fig. 5e1–f3). The length of $CsPbI_3$ nanowire and the incorporation of ionic liquid are both important for realizing bidirectional synaptic photo-response. Supplementary Figs. 6 and 7 show two control devices with longer (~2 μm) $CsPbI_3$ nanowire, and without ionic liquid, respectively. With longer perovskite nanowire, as shown in Supplementary Fig. 6, the positive surrounding gate effect can hardly reach the $CsPbI_3/Pb$ interface and create an electron Schottky barrier as shown in Supplementary Fig. 4d3. Thus, the device only generates positive photocurrent under different color illumination. Also, without the mobile ions in the ionic liquid which suppress the positive surrounding gate of NiO (Supplementary Fig. 7), the device can only generate a negative photocurrent.

Apart from novel bidirectional photo-response behavior, the device also shows good weak light sensitivity and a large dynamic

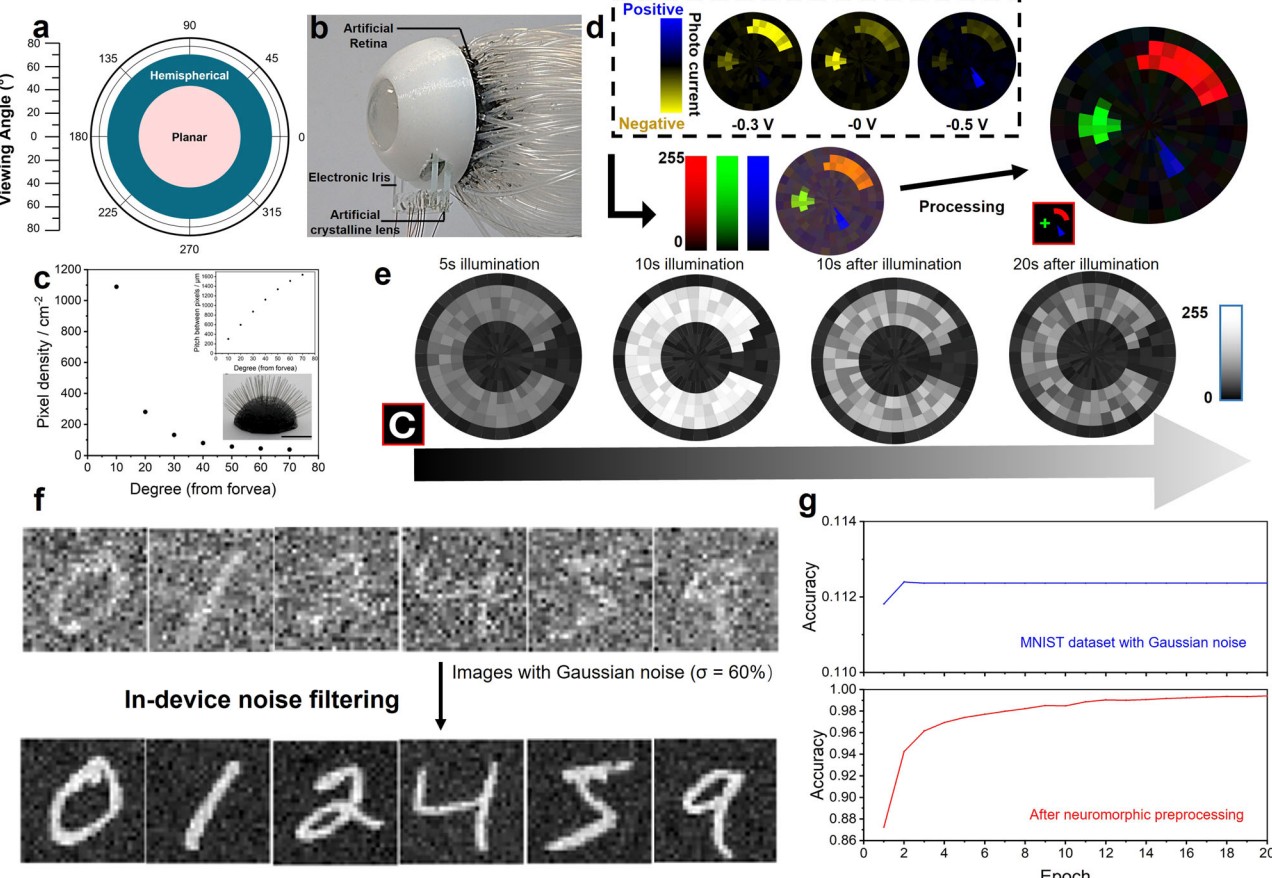

**Fig. 4 | Characterization and neuromorphic image sensing demonstration of the bionic eye. a** Viewing angle of the hemispherical and planar devices. **b** Photo of the bionic eye device. **c** Area dependent pixel density distribution of the artificial retina. The inset figures show the pixels' pitch distribution (up) and a photo of Ni microwire electrodes aligned on the hemispherical retina, with a scale bar of 1 cm (down). **d** Demonstration of color pattern reconstruction. Including three current maps measured with different biases, reconstructed images before and after processing. **e** Demonstration of neuromorphic preprocessing ability: potentiation and depression process of the hemispherical retina with 0 V bias under continuous illumination. **f, g** Demonstration of in-device noise filtering function. **f** Image samples of MNIST dataset with Gaussian noise ($\sigma = 255 \times 60\%$), and reconstructed images with in-device noise filtering. **g** CNN-based pattern recognition accuracy curves of Gaussian noise infested MNIST dataset without and with neuromorphic preprocessing.

range. Supplementary Fig. 8a, b shows the self-powered photocurrent under illuminations with varying light intensity. The device is sensitive to red, green, and blue light down to 400, 107, and 8.59 μW/cm², respectively. And the highest detectable light intensity is >40 mW/cm². Although the operational speed is limited by relative slow recovery process under blue light, we can use an electronic iris to control the optical stimuli that allow the device to achieve higher frame rate. Supplementary Fig. 8b shows the self-powered photocurrent under ~2 Hz stimuli, the device still shows repeatable photoresponse. In addition, due to the excellent packaging of the inert PAM, the device demonstrates excellent stability. Thanks to the good stability of inorganic perovskite[30–33], Supplementary Fig. 9 shows that the device can work for more than four months in an in-door environment.

The unique working principle and the intriguing synaptic behavior of our device enable filter-free color imaging ability that has great potential in machine vision applications. For example, image classification is the basis of computer vision, and it requires efficient color imaging hardware. However, in a single-step measurement, red and green color with different light intensities can hardly be separated. To reach a better color recognition standard, we developed a full-color imaging process based on three-step measurement. Figure 3 shows our color pattern reconstruction and image classification system. To further improve the color distinction ability of the bionic eye, we tuned the color selectivity by applying small external voltages. For example,

small positive bias (<0.5 V, Pb with respect to Au) can enhance positive photocurrent under blue light. Thus, the photocurrent under 0.5 V bias is mainly dominated by blue light positive photocurrent. On the other hand, when a small negative bias (>−0.3 V, Pb with respect to CsPbI₃) is applied, the Schottky barrier for hole transportation at CsPbI₃/Pb interface will be reduced. Then carriers generated by red light with more photons in the same light intensity can flow to external circuit more easily and generate relatively stronger negative photocurrent than green and blue light. Therefore, combining the photo-response of three biasing conditions, namely, negative, zero and positive bias, Red, Green and Blue information can be extracted, and the device can reconstruct color patterns with high fidelity. Figure 3a presents the color pattern reconstruction simulation of Cifar 10 dataset based on the single pixel response. For each figure in the dataset, we simulate the bionic eye imaging based on −0.3 V, 0 V, and 0.5 V biased artificial retina, and collect three photocurrent maps, respectively. (Photocurrent density under different wavelength illumination and certain external biases are shown in Fig. 3b.) Then the three photocurrent maps are normalized to three 8 digits' grayscale patterns, which are directly sent to R, G, and B channels, respectively, to form the reconstructed images. Figure 3a shows examples of original dataset and that reconstructed by the artificial retina, along with the reconstruction of a mono-color image sensor as comparison. Although the color is slightly different from the original dataset, there is enough color information for the following image classification using CNN. Note that biological

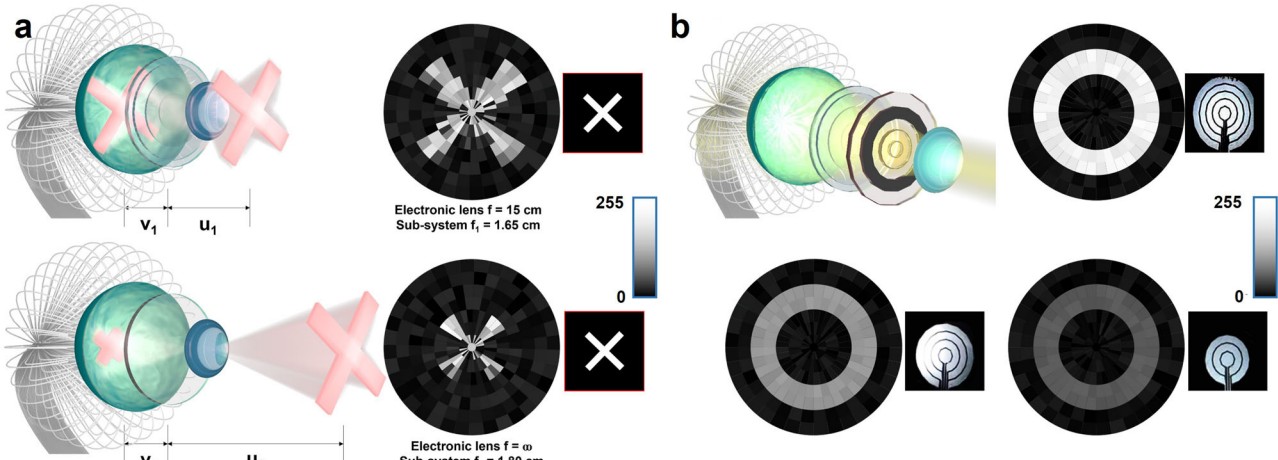

**Fig. 5 | Imaging demonstration along with adaptive optics. a** The schematics and corresponding image detected by the retina under 0 V bias with different focal length controlled by electronics lens. **b** The schematic of assembled bionic eye system and image detected by the retina under 0 V bias with different iris conditions photos of which are inserted beside the corresponding images.

visual perception-inspired CNN is a promising architecture for artificial intelligent (AI) applications such as visual recognition and image classification. Among typical CNNs, residual network (ResNet)[34] is an efficient framework to build a deeper CNN for more complicated AI tasks. Here, we build a typical ResNet 18 architecture with 17 convolutional layers and 1 fully connected layer (Fig. 3c) to demonstrate image classification. The pattern classification accuracies of reconstructed datasets are shown in Fig. 3d. The two identical curves indicate that the color images reconstructed by the artificial retina can be recognized by the CNN with higher accuracy compared to that reconstructed by a conventional mono-color image sensor.

### Hemispherical retina and neuromorphic preprocessing

The hemispherical nanowire array retina, along with its imaging and neuromorphic preprocessing abilities are demonstrated in Fig. 4. Figure 4a shows the FoV comparison between the hemispherical retina and a planar image sensor with the shape of its planar projection. The FoV of our hemispherical device reaches >140° with a single convex lens, much wider than its planar counterpart (~86°). The calculation of FoV is shown in Supplementary Fig. 10. The photograph and schematic structure of the bionic eye is shown in Fig. 4b and Supplementary Fig. 11, respectively. The nickel microneedles are used to emulate visual nerves. Each pixel has a footprint (~$7.85 \times 10^{-5}$ cm$^2$), which is defined by the cross-sectional area of a microneedle. Note that akin to a human eye, the nanowire array retina has photoreceptors with a polar distribution, that is, the photoreceptor density decreases from the fovea to the edge, facilitating excellent balance between pixel number and visual range. Specifically, we adopt a geographic coordinate system (Supplementary Fig. 12) on the hemispherical retina and set the positions of the microwires back contact with a 10° interval either on the latitude (0°-70°) or longitude (0-360°). Supplementary Figs. 13–14 present the simulated detection and related artificial visual recognition of MNIST dataset and Fashion MNIST dataset. Although it can hardly outperform matrix pixel distribution in the reconstruction of non-centered-oriented images, polar pixel distribution with lesser number of pixels can reach recognition accuracy comparable to that with conventional matrix distribution towards centered-oriented images. The area-dependent pixel density is shown in Fig. 4c. Although the nanowire density is high, the resolution of the current device is limited by manual wire bonding process. In the future, high-resolution devices may be fabricated with the assistance of technologies such as high-precision programmable robotic arms, high-resolution 3D printing or laser cutting.

To demonstrate color pattern reconstruction and neuromorphic image sensing functions, we fix the conditions of tunable optics and project different patterns onto the device. Figure 4d shows the color pattern reconstruction ability of the device. Based on the unique color-dependent bidirectional photo-response, the device successfully reconstructed shapes with different colors. In specific, we first applied −0.3 V, 0 V, and 0.5 V bias onto the device and record three photocurrent maps. Then, three maps are directly normalized and sent to R, G and B channels to form a primary image, which has a color difference compared to the original pattern. To achieve a reconstruction with better quality, we calculated the original color pattern based on three current maps. The reconstructed image after processing shows a good fidelity. Figure 4e shows the detection and degradation of a letter "C". The letter attains more clarity with longer illumination time. The result exhibit neuromorphic contrast enhancement ability of the device. Under multiple times of exposure, carriers generated by previous stimuli can be partially stored in the device to enhance lateral response.

The preprocessing ability can improve the recognition accuracy in real-life applications by processes such as in-device noise filtering. To simulate environmental and systematic noise, we add Gaussian noise ($\sigma = 255 \times 60\%$) to standard MNIST dataset. For randomly occurring noise, the potentiation process can enhance the contrast between target pattern and noise signals. Here we demonstrate a 5 s noise filtering under a 100 Hz optical input of images with noise. The comparison of images with Gaussian noise and after neuromorphic preprocessing is shown in Fig.4f. The datasets before and after noise filtering are both sent to the CNN in Fig. 3d for pattern recognition. The accuracy curves shown in Fig. 4g indicate that the CNN can have much higher recognition accuracy after in-device neuromorphic preprocessing (99.4 % compared to 11.2 %, after 20 epochs). The combined results above prove the potential of improving the accuracy and efficiency of artificial color vision by such an artificial retina.

### Imaging demonstration along with adaptive optics

To replicate the excellent optical adaptivity of human eyes, we have built an optical system comprising a fixed lens (artificial cornea), an electronic iris, and an artificial crystalline lens. Both electronic iris and artificial crystalline lens are based on liquid crystal. Related fabrication processes and performance characterizations are shown in Supplementary Figs. 16–18. The artificial crystalline lens is a liquid crystal Pancharatnam-Berry lens[35–39]. The phase distribution of the lens is shown in Supplementary Fig. 19. The artificial crystalline lens shows reduced geometrical aberration compared to

a conventional glass lens (Simulated results are shown in Supplementary Fig. 20). Furthermore, by controlling the electric field, the lens can be quickly switched (within 16 ms) from light refraction mode with focal length of 15 cm to the normally transparent mode. Figure 5a shows the detection of a letter "X" from two object distances. By turning on and off the electronic lens, the bionic eye exhibits multiple focal lengths, with a near point of 25 cm and a far point of infinity. This feature is comparable to that of the human eye (Supplementary Fig. 21). Supplementary Fig. 22 shows the comparation of well-focused and blurry images acquired with different object distances and electronic lens states. The tunable focal length endows the potential in complex vision functions such as depth imaging and 3D vision. The liquid crystal electronic iris consists of five concentric rings within an area of 78.5 mm$^2$. The transparency of each ring can be individually controlled from 0% to 95% and thus the overall aperture size can be tuned from 3.14 mm$^2$ to 78.50 mm$^2$. This enables fast control (within 13 ms) of the amount of light reaching the retina, alike human iris with pupil. Figure 5b shows the detection of a "O" shape with different electronic iris states. Supplementary Fig. 23 shows the related brightness distribution, where pixels detected higher brightness with larger transparency area in the iris. In general, liquid crystal-based adaptive optics can be controlled with electric field. Therefore, they can be faster, more compact and more energy-efficient than conventional mechanically controlled optics. To further improve the adaptivity of the optics, we simulated a three artificial crystalline lenses-based system, which is shown in Supplementary Fig. 24. Such a system can be switched from 8 different focal lengths to obtain a close to continuous focal length change.

## Discussion

Overall, in this work, we designed and implemented a unique hemispherical bionic retina and spherical eye device, which possess the missing capabilities of color vision, optical adaptivity, and energy efficiency in the previous research. The bionic eyes with filter-free color vision, neuromorphic preprocessing, and adaptive optics, present a new paradigm, instilling structural and functional traits akin to the biological eyes, and uncover the promising potential in further optimization and exploration of artificial visual systems with biomimetic functionalities that can further improve the efficiency of machine vision and robotics.

## Methods

### Materials

All the chemicals were purchased from Sigma-Aldrich and used as received without further purification.

### Preparing free-standing hemispherical PAM with Pb nanoclusters

A hemispherical PAM is fabricated on a hemispherical aluminum substrate, which is achieved by deforming a 0.5 mm thick aluminum sheet with the aid of a hemispherical mold. Then we used a two-step hard anodization process to prepare PAM with 40 μm thickness, 130 nm channel diameter, and 350 nm pitch between the nanowires. Both the anodization processes were performed in 0.1 M oxalic acid aqueous solutions. The voltage was gradually increased to 120 V, and then kept at 120 V for 1 hour. In between the two anodization processes, the first anodization layer was etched away in a mixture of phosphoric acid (6 wt% H3PO4 and 1.8 wt% CrO3) at 98 °C for 1 hour. Pb nanoclusters are deposited at the bottom of the PAM channels via a barrier thinning and following electrochemical deposition processes. Barrier thinning is a voltage-ramping-down (from 120 V to 4 V) process that was carried out to thin down the barrier layer at the bottom of PAM nanochannels. Next, the PAM with Pb nanoclusters was lifted off from the aluminum substrate via a wet etch process in FeCl$_3$ solution.

### NiO hole transporting layer deposition

On the free-standing PAM, ~5 nm nickel oxide (NiO) layer was deposited on the sidewall of the PAM channels by ALD method. The oxygen source and Ni precursor are O$_3$ and C$_{14}$H$_{32}$N$_2$O$_2$Ni, respectively. The process was conducted at 250 °C while the C$_{14}$H$_{32}$N$_2$O$_2$Ni was contained at 130 °C and admitted into the ALD reactor with N$_2$ carrier gas (120 sccm). High purity N$_2$ gas (99.9999%) was used as the carrier and purge gas. The NiO ALD pulse sequence followed C$_{14}$H$_{32}$N$_2$O$_2$Ni injection (1500 ms)/wait (10 s)/purge (10 s)/O$_3$ (1500 ms)/wait (10 s)/purge (10 s) steps, and the thickness was controlled by the ALD recipe cycles. NiO was deposited in the PAM channels post 50 cycles of ALD. After the ALD process, the PAM was annealed at 400 °C for 1 hour in high-purity Argon gas.

### CsPbI$_3$ nanowire array growth

CsPbI$_3$ nanowire arrays were grown by a two-zone chemical vapor deposition (CVD) process. The source was prepared by mixing CsI and PbI$_2$ powder with a molar ratio of 3:1 and annealed at 450 °C in high-purity Ar gas (99.999%) for 1 h. The CVD process was conducted in a vacuum environment with a pressure of 0.55 torr. 10 g source was heated to 450 °C and the source vapor was carried by 120 sccm high purity Ar gas (99.999%). The reaction happened at 370 °C for 3 h with 20 min temperature rising time.

### SnO$_2$ electron transporting layer deposition

~5 nm tin oxide (SnO$_2$) layer was also deposited by ALD method. The oxygen source and Sn precursor are O$_3$ and C$_8$H$_{24}$N$_4$Sn, respectively. The C$_8$H$_{24}$N$_4$Sn was contained in a bubbler at 70 °C and admitted into the ALD reactor with N$_2$ carrier gas (20 sccm). High purity N$_2$ gas was used as the carrier and purge gas. The SnO$_2$ ALD pulse sequence followed C$_8$H$_{24}$N$_4$Sn injection (800 ms)/wait (5 s/purge (10 s)/O$_3$ (800 ms)/wait (5 s)/purge (10 s) steps at 250 °C. The thickness was controlled by ALD recipe cycles. 50 cycles SnO$_2$ was deposited on the top of CsPbI$_3$ nanowire arrays and the shell of PAM channels. The retina was aged in vacuum at 300 °C for 1 h annealing. Finally, a 5 min ion milling processes with Ar ion beam energy of 1000 eV was performed on both front and back surfaces of the retina to remove top surface SnO$_2$ and NiO films to avoid cross-talking.

### Au comment electrode deposition

~50 nm Au was deposited on the concave side of the artificial retina via thermal evaporation. The deposition rate was ~1 nm/s.

### Magnetic field aligned Ni wire electrode arrays fabrication

The back contact electrodes of the bionic retina are Ni microwires with 0.1 mm diameter and 2 cm length. A small amount of silver paste was attached to the top of a Ni microwire to improve its electrical contact with the retina. The wires were assembled on the hemispherical retina via a controlled magnetic field. The magnetic field is generated by a spherical permanent magnet and aligned by an iron ball beneath the hemispherical retina. With the magnetic field vertical to the hemispherical surface, Ni wires can vertically stand on the backside of the retina. A fixed red laser was used for marking the electrode position to put on the Ni wire. With a rotating stage, 36 microwire electrodes are assembled along one line of latitude with 10° interval, starting from the 20° latitude. With a lifting platform, 252 electrodes are assembled from 20° to 80° latitude on the hemispherical retina. Together with only one microwire standing at the north pole, there are 253 Ni microwire electrodes in total. After microwire assembling, they were fixed by dropping and curing ultra-violet (UV) epoxy on the array.

### Spherical bionic eye assembling

200 nm ITO was coated on a transparent concave glass as the front common electrode. The hemispherical retina with fixed Ni wire array was packaged on the spacer after injecting ionic liquid in between. The

ITO front electrode was sealed on the retina with cured UV epoxy at the circular edge. The Ni wire electrodes were connected to a PCB board interface through longer copper wires, assisted by a small amount of carbon paste. The whole Ni wire/Cu electrode assembly was wrapped by a plastic tube to avoid contact with other electrode.

## Image sensing and synaptic measurement

The single-color light pulse was generated by lasers. And optical patterns were generated by a projector. The photocurrent was measured by Keithley 2450 source meter unit (SMU). The bionic eye was connected to the SMU through 2 Ni box multiplexer units (PXI2530B, National instruments) which are installed inside of a chassis (PXI1031, National instruments).

## Electronic iris fabrication

A tunable LC iris is fabricated with both variable aperture and efficiency from 0.5 mm to 10 mm in five steps and analog efficiency, respectively. A patterned ITO and another common ITO are placed on the two sides of the LC. The gap between the ring electrode is 50 μm which is small enough when in the near field of the eyes. The LC molecule is aligned vertically in the same direction (88°). When the electric field is applied at different rings on the patterned side, the LC will show spatial rotation and analog efficiency. The LC molecule is a negative liquid crystal and will be perpendicular to the direction of the electric field. Maximum transmittance is achieved when the retardation of the liquid crystal is at half-wave condition. The transmittance is zero when the voltage is smaller than the threshold voltage. With a gradually increasing electric field, the LC molecule will lie down and show transmittance.

## Artificial crystalline lens fabrication

The PB lens is fabricated by coating one glass substrate with SD1 (sulfuric azo-dye) after UV ozone exposure, followed by baking for solvent evaporation. As a photo-rewriteable alignment material, the PB pattern is mapped onto the substrate by interference. Specifically, the alignment pattern is written based on the interference of the blue light (λ = 450 nm) in transmitted mode utilizing Michelson interferometer. Then, assemble the two substrates by blowing spacers. The thickness of the cell is maintained at 3 μm by the glass spacers. Lastly, it is filled with LC, utilizing the capillary phenomenon.

## Data availability

The data that support the findings of this study are provided in the main text and the Supplementary Information. More data are available from the corresponding author upon reasonable request.

## Code availability

The codes used for the simulation are available from the corresponding author on reasonable request.

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

## Acknowledgements

We thank R. Ho, Y. Zhang, and A. H. K. Wong from the Material and Characterization Preparation Facility (MCPF) at HKUST for technically assisting in acquiring TEM images and SEM images. This work was supported by the Science and Technology Plan of Shen Zhen (JCYJ20170818114107730, JCYJ20180306174923335), The General Research Fund (projects 16205321, 16214619) from the Hong Kong Research Grant Council, Innovation Technology Fund (GHP/014/19SZ), Guangdong-Hong Kong-Macao Intelligent Micro-Nano Optoelectronic Technology Joint Laboratory (project 2020B1212030010), and Foshan Innovative and Entrepreneurial Research Team Program (2018IT100031). We also acknowledge the support from the Center for 1D/2D Quantum Materials and the State Key Laboratory of Advanced Displays and Optoelectronics Technologies at HKUST.

## Author contributions

Z.F. and Z.L. conceived the ideas of the work; Z.L., X.Q., C.L. J.C., Z.S., Z.Y., Y. Zhang, Y.D., L.G., Y. Zhou, and S.P. fabricated the device and performed the characterization. Z.L. and W.T. delivered COMSOL simulation. Z.L. and C.L. J.C. developed the CNN and delivered related simulations. Z.L., S.P., A.K.S., C.Y., X.Z. G.S., and Z.F. carried out the data analysis and wrote the manuscript. All authors discussed the results and commented on the manuscript. All data is available in the main text or the Supplementary.

## Competing interests

The authors declare no competing interests.
