## [Peer Review File · Nature Communications]

REVIEWER COMMENTS

Reviewer #1 (Remarks to the Author):

The authors demonstrated a hemispherical bionic eye based on a SnO₂/NiO double-shell nanotube/ a CsPbI₃/NiO core-shell nanowire for achieving color dependent bidirectional synaptic photo-responsive and filter-free colour vision. In addition, tunable liquid crystal optical components that consist of the artificial crystalline lens and electronic iris were used for compact and multi-functional imaging system. Furthermore, neuromorphic pre-processing techniques were adopted based on the unique charge storage and transportation of a metal-oxide nanotube/perovskite nanowire hybrid structure. Generally, the authors demonstrated a novel approach of fabricating artificial imaging devices with intriguing functionalities. Thus, the reviewer recommends the acceptance of this paper by only if comments below could be fully addressed in major revisions.

Comment #1: The authors used the artificial crystalline lens for focal length modulations and electronic iris for dynamic range modulations. Compactness/small form factor are crucial requirements for developing artificial imaging systems, but they should be followed by reliable imaging quality and comparable optical properties to the conventional ones. Therefore, the authors should also offer some optical characteristics compared to conventional lenses (e.g., various coefficient of optical aberrations) to further justify the use of such tunable optical parts for the adaptive optics.

Comment #2: The dynamic range of the image sensor array is also an important factor that determines the dynamic range of the entire system. How sensitive to the light is your image sensor?

Comment #3: In Fig. 5a, the authors showed pixelated images, but it is a bit ambiguous to judge whether they are well-focused. Showing blurred images (i.e., out of focus images) would be helpful. Another suggestion is to provide with some plots displaying various optical parameters (e.g., spot sizes, aberrations) for quantitative analysis.

Comment #4: In Fig. 3d, the authors showed a plot having two identical curves. However, it doesn't look straightforward that it is showing an advantage of using the artificial imaging system for more accurate/efficient image classification tasks.

Comment #5: According to Fig. 4a and SI Fig. 12, imaging demonstrations for Fig. 4d-e seem to be performed using a fixed single convex lens without tunable parts. The authors should explain why they use the fixed system instead of using the tunable system in this section.

Comment #6: In SI Fig. 12, the authors explained that their polar distributed sensors could get a better accuracy even with less pixels. It seems that the polar image sensors have higher resolution on the center part than the outer part. Since it could be that it is only specializing in capturing center-oriented images, showing results with dataset containing non-centered-oriented images (e.g., fashion MNIST dataset) would be helpful to confirm.

Reviewer #2 (Remarks to the Author):

Zhenghao Long and co-authors reported a perovskite nanowire-based bionic eye that mimics the color vision and neuromorphic preprocessing capabilities of the human retina. Artificial retina consisting of perovskite nanowires and SnO₂/NiO₂ metal oxide shells filled with ionic liquids was optimized for wavelength-dependent bidirectional synaptic photoresponses. Furthermore, the color sensitivity could be controlled by adjusting the external bias. Finally, the authors demonstrated that color images reconstructed by the artificial retina could be recognized by convolution neural network with high classification accuracy. In addition, two types of optics, i.e., an artificial crystalline lens and an electronic iris were fabricated and integrated with artificial retina to complete the neuromorphic bionic eye. The functions of both optics were verified through the imaging demonstration. The focal length could be switched by the artificial crystalline lens to detect objects with different distances, and the amount of light reaching the artificial retina could be controlled by the electronic iris. Compared to the previously reported perovskite-based neuromorphic vision systems, the artificial retina consisting of perovskite nanowires had clear advantages in terms of structure (hemispherical structure) and function (filter-free color vision). It was especially impressive that the authors demonstrated the bionic eye by integrating optical components with the artificial retina. This study is expected to contribute to the development of soft bio-inspired electronics field. Therefore, the reviewer recommends publication of this manuscript in Nature communications. Some detailed comments that may help the authors improve this work are as follows.

Comments #1: The authors mentioned that the photocurrents induced by red and green light could be distinguished by their amplitudes under different biases. However, in Fig. 2a and b and Fig. 3b, the photocurrents under red and green illumination were somehow in a similar range (~ tens of $\mu\text{A cm}^{-2}$). The difference of their amplitudes may not be enough to be discriminated. Some comments and/or clarifications are needed.

Comments #2: In Fig. 2 and 3, the authors demonstrated color vision of artificial retina only when the light intensity of each wavelength was the same (11 mW/cm^2). When the light intensity of red light is higher than that of green light, the photocurrent under red illumination is higher than photocurrent under green illumination, so the red and green light cannot be distinguished by the current amplitude. Please explain in detail how to distinguish red and green light if the light intensity of each wavelength is different.

Comments #3: In Fig. 3, the authors simulated the color pattern reconstruction based on the single pixel response. As the authors demonstrated neuromorphic sensing of bionic eye in Fig. 4, the color pattern reconstruction also needs to be simulated based on the nanowire array retina rather than single pixel.

Comments #4: In Introduction, the authors claimed that the modulation of artificial lens and iris by the electric field is more efficient than the conventional mechanically controlled optics. Please provide quantitative comparison for this claim.

Comments #5: In Fig. 5, pattern detection was only demonstrated by the artificial retina with electronic lens, and the function of the electronic iris was demonstrated by the collimated light rather than the pattern. Pattern detection by the assembled bionic eye system should be also demonstrated in order to show that the assembled bionic eye functions normally when the electronic lens and iris are integrated.

Reviewer #3 (Remarks to the Author):

In this work, Long et al. report a very intriguing bionic eye design and implementation. The bionic eye uses a hemispherical nanowire array retina thus has a spherical shape which has a high degree of fidelity as compared to a biological eye. Even though this research group has published an earlier work in Nature in 2020, also based on hemispherical nanowire array, I found the current work has shown a set of major advancement with high novelty, as compared to their earlier work. Specifically, by using a unique heterojunction design using oxides together with perovskite material, very unique color distinction function is realized. This addresses a critical issue of all the bionic eye/retina devices reported earlier. The bipolar spectral dependent photocurrent response mechanism itself is also unique and interesting, it avoids the sophisticated RGB multi-pixel design and simplified the device structure. Of course, the authors mentioned the retina can work under zero bias voltage, this is also meaningful not only to save the device power consumption but also to extend device lifetime. Another novelty of the work is that the authors integrated tunable optics based on two types of liquid crystal devices. Thus the focal length and amount of light reaching the retina can be adjusted. These are all advanced functions which have never been demonstrated by other reported bionic eyes. Overall, I am quite impressed by the unique design, interesting mechanism and powerful functions of the bionic eye here. This type of device can have broad applications in robotics and vision prosthesis. I believe the work is solid, impactful and has high novelty that can certainly meet the high standard of Nature Communications. Therefore, I would like to recommend acceptance if the authors can address the following minor issues:

1. The authors claim the device can operate under zero bias in the beginning, later different bias voltages are used for color distinction. So is the “self-power” mode for monochrome imaging? Please clarify this.
2. For color distinction, I noticed the authors used quite low bias, from -0.5V to +0.5V. Why bias voltages are so small? Shouldn't higher voltage provide more photocurrent gain? Do authors want to obtain low power operation thus low voltages are used? I wonder what will happen if a high bias voltage such as 3~5V is used.
3. What is the weak light detection capability of the retina device? Please show the power dependent measurement results.
4. It seems the authors use multiple measurements with different bias to acquire the color information. In certain way, it is using the response speed to trade with the spectral response. So eventually, what is the response speed for monochrome detection and full color detection? Please supplement with necessary data.
5. A related question to #4, the erasing of synaptic signal requires some time, in practice, how to use the device in high frequency?
6. Interestingly, the authors used liquid crystal lens to tune the focal point. However, the current version device only has a finite number of focal points which is less practical. So how to achieve close to continuous adjustment of focal length with liquid crystal? It will be quite meaningful. Also, what is power consumption of focal point switch? I hope the power is much lower than a mechanical-optical system.
7. There are limited pixels (~250) currently, I noticed the authors use a common front electrode and 250 individual back electrodes. I understand it is extremely challenging to fabricate contacts on a non-planar substrate. Nonetheless this is not an efficient way to achieve high pixel numbers. I would suggest the authors at least discuss about other potential alternative methods to implement high density contacts to make much more pixel numbers for future applications.

*Response letter to reviewers' comments on manuscript: Nature Communications
NCOMMS-22-49249A*

A neuromorphic bionic eye with filter-free colour vision using hemispherical perovskite nanowire array retina

The authors wish to acknowledge the reviewers for providing constructive comments. We have found that the comments were highly valuable for us on improving the quality of the article, thereby making it technically and scientifically more sound. In the past **2 Months**, we have worked extremely hard to address all the points raised by the reviewers to the best of our capability. The reviewers can find the point-to-point response for their critical comments below. All the corresponding changes in the main text and supplementary have been corroborated here and highlighted in the main text and supplementary in red colour.

Reviewer #1 (Remarks to the Author):

The authors demonstrated a hemispherical bionic eye based on a SnO₂/NiO double-shell nanotube/ a CsPbI₃/NiO core-shell nanowire for achieving color dependent bidirectional synaptic photo-responsive and filter-free colour vision. In addition, tunable liquid crystal optical components that consist of the artificial crystalline lens and electronic iris were used for compact and multi-functional imaging system. Furthermore, neuromorphic pre-processing techniques were adopted based on the unique charge storage and transportation of a metal-oxide nanotube/perovskite nanowire hybrid structure. Generally, the authors demonstrated a novel approach of fabricating artificial imaging devices with intriguing functionalities. Thus, the reviewer recommends the acceptance of this paper by only if comments below could be fully addressed in major revisions.

Response: We express our gratitude to the reviewer for appreciating our work, highlighting the novelty and providing critical comments for us to address. Below please find our point-to-point response to the previous questions raised by this reviewer which we also found very helpful on improving our work substantially.

Comment #1: The authors used the artificial crystalline lens for focal length modulations and electronic iris for dynamic range modulations. Compactness/small form factor are crucial requirements for developing artificial imaging systems, but they should be followed by reliable imaging quality and comparable optical properties to the conventional ones. Therefore, the authors should also offer some optical characteristics compared to conventional lenses (e.g., various coefficient of optical aberrations) to further justify the use of such tunable optical parts for the adaptive optics.

Response: We thank the reviewer for this valuable comment. We agree that the comparison of our tunable lens and conventional lens can help to support the advancement of our system. To address this comment, we simulated the optical properties of our artificial crystalline lens and a conventional glass lens. The result can be found in **Fig. R1-1** and **Supplementary Fig. 20**. To properly investigate the performance of the lenses, we simulated both normal and oblique incident illuminations. The modulation transfer function (MTF) charts (Fig. R1-1 a2 and b2) show that artificial lens can have larger MTF under higher cycles/mm (i.e. spatial resolution). The field curvature and distortion curves show (Fig. R1-1 a3 and b3) that the artificial

crystalline lens can provide less field curvature and comparable distortion. The focal points distribution patterns and irradiance curves (Fig. R1-1 a4-5 and b4-5) show that the artificial crystalline lens can focus light to a smaller region compared to the conventional glass lens. In general, because the artificial crystalline lens focus light via continuous phase change (Supplementary Fig.19), it is free from spherical aberration. The tunable lens can fully correct spherical aberration in normal incident situations and reduce coma aberration in oblique incident situations. We have added the following discussions: “The artificial crystalline lens shows reduced geometrical aberration compared to a conventional glass lens (Simulated results are shown in Supplementary Fig. 20).”

Fig. R1-1 | Optical properties of artificial crystalline lens and conventional glass lens
 Optical simulation of **a1-a5**, artificial crystalline lens and **b1-b5**, a conventional glass lens. **a1, b1**, the schematic of optical path in the simulation. **a2, b2**, MTF – special frequency curves. **a3, b3**, Field curvature and distortion curves. **a4, b4**, focal points distribution patterns. **a5, b5**, spatial distribution of irradiance close to focal point.

Comment #2: The dynamic range of the image sensor array is also an important factor that determines the dynamic range of the entire system. How sensitive to the light is your image sensor?

Response: We thank the reviewer for this valuable comment. We agree with the reviewer that the dynamic range is an important factor and related data should be provided. Therefore, we have measured self-powered photo-response of single pixel under red, green and blue colour illuminations with varying light intensity. As suggested in **Figure R1-2**, the lowest detectable

light intensity for red, green and blue illuminations are 400, 107, and 8.59 $\mu\text{W}/\text{cm}^2$, respectively. And the device is sensitive to illuminations $>40 \text{ mW}/\text{cm}^2$. Limited by our laser, we haven't measured the photo response under stronger illuminations. **Fig. R1-2** shows the photo current under different colour light with multiple light intensity. The result is also shown in Supplementary Fig. 8a-b. In the manuscript, we added the following statement: "Apart from novel bidirectional photo response behavior, the device also shows good weak light sensitivity and a large dynamic range. Supplementary Fig. 8a-b shows the self-powered photo current under illuminations with varying light intensity. The device is sensitive to red, green and blue light down to 400, 107, and 8.59 $\mu\text{W}/\text{cm}^2$, respectively. And the highest detectable light intensity is $> 40 \text{ mW}/\text{cm}^2$."

Fig. R1-2 | **a-b**, power density dependent self-powered photo response in **a**, linear and **b**, log scale.

Comment #3: In Fig. 5a, the authors showed pixelated images, but it is a bit ambiguous to judge whether they are well-focused. Showing blurred images (i.e., out of focus images) would be helpful. Another suggestion is to provide with some plots displaying various optical parameters (e.g., spot sizes, aberrations) for quantitative analysis.

Response: We thank the reviewer for the valuable suggestions. Showing comparison results and related parameters can greatly improve the understanding of related parts. Following the reviewer's suggestion, we have added out of focus images in Supplementary Fig.22 (also shown in **Fig. R1-3**). However, for the high pixel density wire bonding on hemispherical detector is challenging, we can hardly read out the spot size from the image. For comparison, we counted the frequency of pixels under certain brightness. Well focused images (Fig. R1-3

a1-a2 and c1-c2) have separated brightness distribution of bright and dark pixels. We added the following statement to the manuscript: “Supplementary Fig. 22 shows the comparison of well-focused and blurry images acquired with different object distances and electronic lens states.”

Fig. R1-3 | The comparison of well-focused and blurry images.

Well focused and blurry images of **a1-b1**, closed object and **c1-d1**, far away object. And **a2-d2**, related brightness distribution

Comment #4: In Fig. 3d, the authors showed a plot having two identical curves. However, it doesn't look straightforward that it is showing an advantage of using the artificial imaging system for more accurate/efficient image classification tasks.

Response: We thank the reviewer for the valuable suggestions. In the previous manuscript we demonstrated that the simulated detection result of our device can be recognized by convolutional neuron network as accurately as the original RGB images so that our system can have reliable full colour imaging ability. However, the identical curves haven't shown our advantages straightforwardly. Therefore, we delivered another visual classification simulation, in which we compared the classification efficiency based on simulated results of our full colour and a mono colour imaging systems. Overall, the CNN can achieve higher classification accuracy towards colour images reconstructed by the bionic eye. We also modified our discussions as follow: “The two identical curves indicate that the colour images reconstructed

by the artificial retina can be recognized by the CNN with higher accuracy compared to that reconstructed by a conventional mono color image sensor.”

Fig. R1-4 | Artificial colour vision system and its colour pattern recognition performance.

a, The comparison of original objects, and that reconstructed by a grayscale image sensor and our colour sensitive bionic eye devices. **b**, Photo current density of a pixel on the artificial retina with different bias under red, green and blue illuminations. **c**, The architecture of the neural network used in colour pattern recognition. And **d**, identical curves of classification accuracy after several epoch.

Comment #5: According to Fig. 4a and SI Fig. 12, imaging demonstrations for Fig. 4d-e seem to be performed using a fixed single convex lens without tunable parts. The authors should explain why they use the fixed system instead of using the tunable system in this section.

Response: We thank the reviewer for the valuable suggestions. In this part, we demonstrated the neuromorphic imaging ability of the artificial retina, with fixed conditions of tunable optics. Following the reviewer’s suggestion, we have now added more clarification in the revised version: “To demonstrate color pattern reconstruction and neuromorphic image sensing

functions, we fixed the conditions of tunable optics and projecting different pattern onto the device.”

Comment #6: In SI Fig. 12, the authors explained that their polar distributed sensors could get a better accuracy even with less pixels. It seems that the polar image sensors have higher resolution on the center part than the outer part. Since it could that it is only specializing in capturing center-oriented images, showing results with dataset containing non-centered-oriented images (e.g., fashion MNIST dataset) would be helpful to confirm.

Response: We thank the reviewer for the valuable suggestions. We agree that adding related result based on non-centered-oriented images can help to study the advantages and limitations of polar pixel alignment. Following this comment, we prepared imaging simulation based on fashion MNIST dataset (**Fig. R1-5** and Supplementary Fig. 14). We not only simulated polar distribution with 7×36 pixels like our device, but 20×36 pixels (comparable to 28×28) as well. The result shows that polar pixel distribution has worse accuracy to non-centered-oriented images. Therefore, we modified the claim as follow: “Although it can hardly outperform matrix pixel distribution in the reconstruction of non-centered-oriented images, polar pixel distribution with lesser pixels can reach recognition accuracy comparable to that with conventional matrix distribution towards centered-oriented images.”

Fig. R1-5 | Non-centered-oriented images reconstruction and related pattern recognition of image sensors with matrix and polar distribution

a. Images reconstruction based on matrix and polar pixel distributions. And b. related identical curves of recognition accuracy after several epochs.

Reviewer #2 (Remarks to the Author):

Zhenghao Long and co-authors reported a perovskite nanowire-based bionic eye that mimics the color vision and neuromorphic preprocessing capabilities of the human retina. Artificial retina consisting of perovskite nanowires and SnO₂/NiO₂ metal oxide shells filled with ionic liquids was optimized for wavelength-dependent bidirectional synaptic photoresponses. Furthermore, the color sensitivity could be controlled by adjusting the external bias. Finally, the authors demonstrated that color images reconstructed by the artificial retina could be recognized by convolution neural network with high classification accuracy. In addition, two types of optics, i.e., an artificial crystalline lens and an electronic iris were fabricated and integrated with artificial retina to complete the neuromorphic bionic eye. The functions of both optics were verified through the imaging demonstration. The focal length could be switched by the artificial crystalline lens to detect objects with different distances, and the amount of light reaching the artificial retina could be controlled by the electronic iris. Compared to the previously reported perovskite-based neuromorphic vision systems, the artificial retina consisting of perovskite nanowires had clear advantages in terms of structure (hemispherical structure) and function (filter-free color vision). It was especially impressive that the authors demonstrated the bionic eye by integrating optical components with the artificial retina. This study is expected to contribute to the development of soft bio-inspired electronics field. Therefore, the reviewer recommends publication of this manuscript in Nature communications. Some detailed comments that may help the authors improve this work are as follows.

Response: We express our great gratitude to the reviewer for appreciating our work, highlighting the novelty and providing critical comments for us to address. Below please find our point-to-point response to the questions and comments raised by this reviewer which we also found very helpful in improving our work substantially.

Comments #1: The authors mentioned that the photocurrents induced by red and green light could be distinguished by their amplitudes under different biases. However, in Fig. 2a and b and Fig. 3b, the photocurrents under red and green illumination were somehow in a similar range (~ tens of $\mu\text{A cm}^{-2}$). The difference of their amplitudes may not be enough to be discriminated. Some comments and/or clarifications are needed.

Response: We thank the reviewer for the careful observation. In our previous manuscript, we demonstrated the colour dependent bidirectional photo response. We agree that the bidirectional colour response can only efficiently distinguish blue illumination from red and green illumination. In single step measurement, red and green illuminations with higher and lower intensity, respectively, can hardly be distinguished. To acquire full colour images, we demonstrated a three-step measurement based full colour imaging process. By modifying colour selectivity with low biases, red and green light can be distinguished via multiple measurements. As a further clarification we have now added this statement to the manuscript: “However, in single step measurement, red and green colour with different light intensity can hardly be separated. To obtain a better colour recognition ability, we developed a full colour imaging process based on the three-step measurement.”

Comments #2: In Fig. 2 and 3, the authors demonstrated color vision of artificial retina only when the light intensity of each wavelength was the same (11 mW/cm²). When the light intensity of red light is higher than that of green light, the photocurrent under red illumination is higher than photocurrent under green illumination, so the red and green light cannot be distinguished by the current amplitude. Please explain in detail how to distinguish red and green light if the light intensity of each wavelength is different.

Response: We thank the reviewer for the valuable comment. We agree that in single step measurement, red and green light can hardly be distinguished. Therefore, we demonstrated a three-step measurement based colour imaging process in Fig. 3-4. For example, the device is relatively more sensitive to red light than green light under -0.3 V bias, and is relatively more sensitive to green light than red light under 0 V bias. So, after two-step measurement we can distinguish whether the unknown illumination is red or green with certain light intensity. With three-step measurement as shown in Fig.3-4, the device can reconstruct colour patterns without any further processing.

Comments #3: In Fig. 3, the authors simulated the color pattern reconstruction based on the single pixel response. As the authors demonstrated neuromorphic sensing of bionic eye in Fig. 4, the color pattern reconstruction also needs to be simulated based on the nanowire array retina rather than single pixel.

Response: We thank the reviewer for the valuable suggestion. We agree that colour pattern reconstruction demonstration is necessary. Following this suggestion, we demonstrated a full colour pattern reconstruction experimentally in Fig. 4d. And it is also shown in **Fig. R2-1**. We also added following explanations: “To demonstrate color pattern reconstruction and neuromorphic image sensing functions, we fix the conditions of tunable optics and project different patterns onto the device. Fig. 4d shows the color pattern reconstruction ability of the device. Based on the unique color dependent bidirectional photo response, the device successfully reconstructed shapes with different colors. In specific, we firstly applied -0.3 V, 0 V and 0.5 V bias onto the device and record three photo current maps. Then, three maps are directly normalized and send to R, G and B channels to form a primary image, which has a colour difference compared to the original pattern. To achieve a reconstruction with better quality, we calculated the original colour pattern based on three current maps. The reconstructed image after processing shows a good fidelity.”

Fig. R2-1 | colour pattern reconstruction of the artificial retina

Comments #4: In Introduction, the authors claimed that the modulation of artificial lens and iris by the electric field is more efficient than the conventional mechanically controlled optics. Please provide quantitative comparison for this claim.

Response: We thank the reviewer for the valuable suggestion. Following this suggestion, we compared the efficiency, operational speed, and energy consumption between novel electronic optics and conventional mechanical systems in supplementary Table 1 and **Table R2-1**. Our system achieved a wider field of view despite being smaller in size and having lighter weight. And the system can achieve faster switching speed, and the electrical tunable property. We mentioned “Supplementary Table 1 compares our system with some commercial zoom lenses.” In our revised manuscript.

Table R2-1 | Comparison of commercial and our tunable lens systems

Tunable lens systems	Number of lenses	Diameter and length / mm	Weight / g	Focal length range / mm	FoV	Switch type	Switch time
Our system	2	$\Phi 10 \times 10$	<20	25 & ∞	>140°	Electrical	~5 ms
Canon RF100-500mm F4.5-7.1 L IS USM	20	$\Phi 93.8 \times 207.6$	1370	100-500	24°	Mechanical	>100 ms
NIKKOR Z 100-400MM F/4.5-5.6 VR S	25	$\Phi 98 \times 222$	1355	100-400	24°	Mechanical	>100 ms
FUJINON GF20-35mmF4 R WR	14	$\Phi 88.5 \times 112.5$	725	20-35	108°	Mechanical	>100 ms
Sony FE PZ 16-35mm F4 G	13	$\Phi 80.5 \times 88.1$	353	16-35	107°	Mechanical	>100 ms

Comments #5: In Fig. 5, pattern detection was only demonstrated by the artificial retina with electronic lens, and the function of the electronic iris was demonstrated by the collimated light rather than the pattern. Pattern detection by the assembled bionic eye system should be also demonstrated in order to show that the assembled bionic eye functions normally when the electronic lens and iris are integrated.

Response: We thank the reviewer for the valuable suggestion. We agree that pattern reconstruction demonstration can help to display the function of the electronic iris. We re-demonstrated this part via reconstruction of a “O” shape. The result can be found in **Fig. R2-2** (also in Fig. 5b and Supplementary Fig. 23). With larger transparent size in the iris, the reconstructed “O” is brighter. We also counted the brightness distribution of the results to support the above statement. In Fig. R2-2 a2-c2, pixels distributed in brighter regions when the larger transparent size of electronic iris is larger. We mentioned: “Fig. 5b shows the detection of a “O” shape with different electronic iris states. Supplementary Fig. 23 shows the related brightness distribution, where pixels detected higher brightness with larger transparency area in the iris.” In our revised manuscript.

Fig. R2-2 | Pattern reconstruction with different electronic iris states.

a1-c1. “O” shape reconstruction with different electronic iris state and a2-c2. related brightness distribution.

Reviewer #3 (Remarks to the Author):

In this work, Long et al. report a very intriguing bionic eye design and implementation. The bionic eye uses a hemispherical nanowire array retina thus has a spherical shape which has a high degree of fidelity as compared to a biological eye. Even though this research group has published an earlier work in Nature in 2020, also based on hemispherical nanowire array, I found the current work has shown a set of major advancement with high novelty, as compared to their earlier work. Specifically, by using a unique heterojunction design using oxides together with perovskite material, very unique color distinction function is realized. This addresses a critical issue of all the bionic eye/retina devices reported earlier. The bipolar spectral dependent photocurrent response mechanism itself is also unique and interesting, it avoids the sophisticated RGB multi-pixel design and simplified the device structure. Of course, the authors mentioned the retina can work under zero bias voltage, this is also meaningful not only to save the device power consumption but also to extend device lifetime. Another novelty of the work is that the authors integrated tunable optics based on two types of liquid crystal devices. Thus the focal length and amount of light reaching the retina can be adjusted. These are all advanced functions which have never been demonstrated by other reported bionic eyes. Overall, I am quite impressed by the unique design, interesting mechanism and powerful functions of the bionic eye here. This type of device can have broad applications in robotics and vision prosthesis. I believe the work is solid, impactful and has high novelty that can certainly meet the high standard of Nature Communications. Therefore, I would like to recommend acceptance if the authors can address the following minor issues:

Response: We express our great gratitude to the reviewer for appreciating our work, highlighting the novelty and providing critical comments for us to address. Below please find our point-to-point response to the previous questions raised by this reviewer which we also found very helpful in improving our work substantially.

1. The authors claim the device can operate under zero bias in the beginning, later different bias voltages are used for color distinction. So is the “self-power” mode for monocular imaging? Please clarify this.

Response: We thank the reviewer for the valuable comment. Under “self-power” mode, the device exhibits unique colour dependent bidirectional photo response. The photo current under blue light has a different direction compared to that under red and green light. Therefore, the device can distinguish blue colour from red and green colour. However, it can hardly distinguish red and green colour with different light intensity. So we developed a three-step measurement process to accurately distinguish red, green and blue colors. “Self-power mode” can support a two colour imaging, and multiple measurements with applying different biases can support full colour imaging. To further clarify, we have now added this statement: “However, in single step measurement, red and green colour with different light intensity can hardly been separated. To reach a better colour recognition ability, we developed a full colour imaging process based on three-step measurement.”

2. For color distinction, I noticed the authors used quite low bias, from -0.5V to +0.5V. Why bias voltages are so small? Shouldn't higher voltage provide more photocurrent gain? Do authors want to obtain low power operation thus low voltages are used? I wonder what will happen if a high bias voltage such as 3~5V is used.

Response: We thank the reviewer for the valuable comment. We agree that higher bias can provide more photo current gain. However, with applying higher bias, the motion of carriers will be mainly controlled by external instead of internal electric field. As a result, the relative colour selectivity under high bias is low. So that we can hardly obtain colour information. Therefore, we use lower bias to obtain the colour imaging function. Following this comment, we measured the photo response under 3 V and 5 V biases, the result is shown in **Fig. R3-1**. The device is always more sensitive to shorter wavelength illumination under different biases. And the device can only provide unipolar photo current under higher bias.

Fig. R3-1 | Photo response under higher biases

3. *What is the weak light detection capability of the retina device? Please show the power dependent measurement results.*

Response: We thank the reviewer for the valuable suggestion. We measured the self-powered photo response under different colour illumination with multiple light intensity. The result is displayed in **Fig. R3-2** and Supplementary Fig. 8a-b. the lowest detectable light intensities of the retina device are 400, 107, and 8.59 $\mu\text{W}/\text{cm}^2$ for red, green and blue illumination, respectively. And the highest detectable illumination is stronger than 40 mW/cm^2 . Limited by the light source, we haven't measured photo response under stronger light. In the manuscript, we added the following statement: "Apart from novel bidirectional photo response behavior, the device also shows good weak light sensitivity and a large dynamic range. Supplementary Fig. 8a-b shows the self-powered photo current under illuminations with varying light intensity. The device is sensitive to red, green and blue light down to 400, 107, and 8.59 $\mu\text{W}/\text{cm}^2$, respectively. And the highest detectable light intensity is $> 40 \text{ mW}/\text{cm}^2$."

Fig. R3-2 | a-b, power density dependent self-powered photo response in **a**, linear and **b**, log scale.

4. It seems the authors use multiple measurements with different bias to acquire the color information. In certain way, it is using the response speed to trade with the spectral response. So eventually, what is the response speed for monochrome detection and full color detection? Please supplement with necessary data.

Response: We thank the reviewer for the valuable comment. We agree that the response speed is important for an imaging system. However, it is hard to define the response speed for our synaptic device. In previous demonstration, we used 1 s exposure time, and the full colour frame rate is >9 s considering the recovery process. To show that our device can work at a faster speed, we demonstrated the device response under ~ 2 Hz optical stimuli. As shown in **Fig. R3-3** and Supplementary Fig. 8c, the device shows repeatable photo response. In such a case, a mono colour imaging can be acquired in ~ 0.5 s, and a full color imaging can be acquired in ~ 1.5 s. We also added the following sentences to the manuscript: “Although the operational speed is limited by relative slow recovery process under blue light, we can use electronic iris to control the optical stimuli that allow the device to achieve higher frame rate. Supplementary Fig. 8b shows the self-powered photo current under ~ 2 Hz stimuli, the device still shows repeatable photo-response.”

Fig. R3-3 | Photo response under ~ 2 Hz optical stimuli.

5. A related question to #4, the erasing of synaptic signal requires some time, in practice, how to use the device in high frequency?

Response: We thank the reviewer for the valuable comment. As mentioned in the response to question #4, the device can work at 2 Hz, which is fast enough in some applications such as low speed photography. For high frequency applications, although the device can hardly reach a high frame rate, the synaptic behavior allows it to provide the preprocessed result. For example, the trajectory of a fast-moving object can be shown in one frame to achieve moving object detection, as shown in **Fig. R3-4**. In this way, the device can detect moving objects.

Fig. R3-4 | Trajectory imaging of a moving light spot with two frames

6. Interestingly, the authors used liquid crystal lens to tune the focal point. However, the current version device only has a finite number of focal points which is less practical. So how to achieve close to continuous adjustment of focal length with liquid crystal? It will be quite meaningful. Also, what is power consumption of focal point switch? I hope the power is much lower than a mechanical-optical system.

Response: We thank the reviewer for the valuable comment. We agree that continuous adjustment of focal length can greatly improve the usability of our device. As the electric lens can only be switched binarily, the system can hardly have continuous focal length change like conventional zoom lenses. To achieve close to continuous change of focal length, we simulated an optical system containing three electric lenses. **Fig. R3-5** and Supplementary Fig. 24 shows the schematic of such a system. The combination of three lenses can provide 8 different focal points. And more focal points can be achieved with more electronic lenses. The energy consumption of the focal point switching is 0.1 μJ per switch. In the revised manuscript, we added: “To further improve the adaptivity of the optics, we simulated a three artificial crystalline lenses based system, which is shown in Supplementary Fig. 24. Such a system can be switched from 8 different focal lengths to obtain a close to continuous focal length change.”

State of each lens			System focal length
f_{c1}	f_{c2}	f_{c3}	
on	off	off	f_1
on	off	on	f_2
on	on	off	f_3
on	on	on	f_4
off	off	on	f_5
off	on	off	f_6
off	on	on	f_7
off	off	off	f_8

Fig. R3-5 | Three electronic lenses based imaging system with close to continuous focal length switching.

The schematics of **a**, An electronic lens at “ON” state. **b**, An electronic lens at “OFF” state. And **c**, a 3 electronic lenses based optical system with 8 focal points.

7. *There are limited pixels (~250) currently, I noticed the authors use a common front electrode and 250 individual back electrodes. I understand it is extremely challenging to fabricate contacts on a non-planar substrate. Nonetheless this is not an efficient way to achieve high pixel numbers. I would suggest the authors at least discuss about other potential alternative methods the implement high density contacts to make much more pixel numbers for future applications.*

Response: We thank the reviewer for the valuable comment. We agree that the manual wire bonding process can hardly reach a high resolution. Apart from our current methods, we think the following approaches may help us to fabricate high resolution hemispherical image sensors.

-Wire bonding with robotic arm: In our previous work (Ref. #4, *Nature* **581**, 278–282 (2020)), we demonstrated a Ni micro needle-based wire bonding process that can fabricate pixels with small size ($\sim 1\mu\text{m}$) close to optical diffraction limit. With the aid of high-precision programmable robotic arms in the future, the pixel density of such a device can be improved for commercial applications.

-3D printing mask assisted electrode patterning: The resolution of 3D printing can achieve $<10\mu\text{m}$. In the future, this technology can help us to fabricate high resolution image sensors on hemispherical geometry. For example, parallel ITO lines can be sputtered on both sides of the artificial retina with 3D printed hemispherical shadow masks to form a cross bar structure.

-Laser assisted electrode patterning: Lasers that can move in 3D can also help us in high density electrode fabrication. For example, ITO can be sputtered on each side of the artificial retina, then unwanted parts can be removed by well-controlled laser cutting.

We have added the following discussions to the manuscript: “Although the nanowire density is high, the resolution of the current device is limited by manual wire bonding process. In the future, high resolution devices may be fabricated with the assisted of technologies such as high-precision programmable robotic arms, high resolution 3D printing or laser cutting.”

REVIEWERS' COMMENTS

Reviewer #1 (Remarks to the Author):

I felt that this manuscript was quite strong in its original form. In revisions, the authors have done extensive additional experimental and modeling work, to significantly improve the paper further. I believe that the updated version is suitable for publication, without need for further revisions.

Reviewer #2 (Remarks to the Author):

All comments from the reviewer were addressed well in the revised manuscript which is now ready for publication.

Reviewer #3 (Remarks to the Author):

the authors had addressed all the reviewer's concern.